# COMPRESSION WITHOUT QUANTIZATION

## ABSTRACT

Standard compression algorithms work by mapping an image to discrete code using an encoder from which the original image can be reconstructed through a decoder. This process, due to the quantization step, is inherently non-differentiable so these algorithms must rely on approximate methods to train the encoder and decoder end-to-end. In this paper, we present an innovative framework for lossy image compression which is able to circumvent the quantization step by relying on a non-deterministic compression codec. The decoder maps the input image to a distribution in continuous space from which a sample can be encoded with expected code length being the relative entropy to the encoding distribution, i.e. it is bits-back efficient. The result is a principled, end-to-end differentiable compression framework that can be straight-forwardly trained using standard gradient-based optimizers. To showcase the efficiency of our method, we apply it to lossy image compression by training Probabilistic Ladder Networks (PLNs) on the CLIC 2018 dataset and show that their rate-distortion curves on the Kodak dataset are competitive with the state-of-the-art on low bitrates.

## 1 INTRODUCTION

The recent development of powerful generative models, such as Variational Auto-Encoders (VAEs) and their hierarchical extensions, such as Probabilistic Ladder Networks (PLNs) (Kingma & Welling, 2014; Sønderby et al., 2016; Higgins et al., 2017) has caused a great deal of interest in their application to lossy compression, notably Ballé et al. (2016); Theis et al. (2017); Rippel & Bourdev (2017); Ballé et al. (2018); Mentzer et al. (2018); Johnston et al. (2018). The benefit of using these models as opposed to hand-crafted methods is that they can adapt to the statistics of their inputs much better, and hence allow significant gains in compression rate at a given quality setting, and in quality at a given compression rate. A second advantage is their easier adaptability to new media formats, such as light-field cameras, 360° images, Virtual Reality (VR), video streaming, etc. for which classical methods are not currently applicable, or would perform pathologically badly.

Lossy compression codecs usually perform a lossy transformation on the continuous representation of the data, after which it is mapped to a discrete domain through an operation known as quantization, such that entropy coding may be used to encode this transformed representation. This continuous-to-discrete mapping will mean that quantization will necessarily have zero derivative almost everywhere. Since the above mentioned generative models are usually trained using some gradient-based technique, this means that they cannot be directly applied as compression models, as quantization will destroy the learning signal, and hence a workaround is needed. A nice remedy to this problem is to replace the quantities affected by quantization (or at least their derivatives) by some smooth approximation during training, as is done in recent approaches, e.g. Ballé et al. (2016); Theis et al. (2017); Ballé et al. (2018); Mentzer et al. (2018); Johnston et al. (2018).

Similarly to Ballé et al. (2016); Theis et al. (2017); Ballé et al. (2018), we aim to minimize the rate-distortion of our model directly, but unlike them we replace entropy coding with a more general coding technique that allows us to use probability densities for coding instead of probability masses. This then allows us to forgo the quantization operation altogether and we get an end-to-end differentiable model that can be optimized with standard gradient-based optimizers.

We select an appropriate generative model $p(\mathbf{x}, \mathbf{z}) = p(\mathbf{x} \,|\, \mathbf{z})p(\mathbf{z})$ with an approximate posterior $q(\mathbf{z} \,|\, \mathbf{x})$, where the latents $\mathbf{z}$ are going to serve as the transformed representation of $\mathbf{x}$ that we wish to compress. We show that $-\log p(\mathbf{x} \,|\, \mathbf{z})$ corresponds to the distortion and $\mathrm{KL}\left[\, q(\mathbf{z} \,|\, \mathbf{x}) \,||\, p(\mathbf{z}) \,\right]$

corresponds to the rate of the model. Hence, optimizing our model for the rate-distortion is as simple as minimizing $-\log p(\mathbf{x} \mid \mathbf{z}) + \beta \mathrm{KL}\left[\, q(\mathbf{z} \mid \mathbf{x}) \, || \, p(\mathbf{z}) \,\right]$, where $\beta$ is a scalar hyperparameter controlling the compression rate.

To compress some data $\boldsymbol{x}$ with our trained model, instead of selecting $\boldsymbol{z}$ deterministically and then coding it, we map $\boldsymbol{x}$ to the posterior $q(\mathbf{z} \mid \boldsymbol{x})$ and use it to code a sample from it, using an adapted version of the importance sampling algorithm proposed by Havasi et al. (2018). First, set $K = \mathrm{KL}\left[\, q(\mathbf{z} \mid \boldsymbol{x}) \, || \, p(\mathbf{z}) \,\right]$ and stochastically draw independent samples $(\boldsymbol{z}_1, \ldots, \boldsymbol{z}_{\lceil \exp\{K\} \rceil})$ from $p(\mathbf{z})$. Next, select $\boldsymbol{z}_{c^*}$ for some $1 \leq c^* \leq \exp\{K\}$. We can use the *index* $c^*$ as the compressed representation of $\boldsymbol{z}_{c^*}$. Clearly, $c^*$ can be coded in $K$ nats, and it can also be shown that $\boldsymbol{z}_{c^*}$ will be a low-bias sample from $q(\mathbf{z} \mid \boldsymbol{x})$. As the efficiency of this algorithm is $\mathrm{KL}\left[\, q(\mathbf{z} \mid \boldsymbol{x}) \, || \, p(\mathbf{z}) \,\right]$, we will refer to it as relative entropy coding (REC), and note that REC algorithms achieve bits-back efficiency (Hinton & Van Camp, 1993).

The contributions of this paper are as follows:

- We show how any generative model $p(\mathbf{x}, \mathbf{z}) = p(\mathbf{x} \mid \mathbf{z})p(\mathbf{z})$ with approximate latent posterior $q(\mathbf{z} \mid \mathbf{x})$ can be used for bits-back efficient lossy transform coding, using an adapted version of Minimal Random Code Learning (MIRACLE, Havasi et al. (2018)).

- We present the concept of *relative entropy coding* (REC) algorithms and attainable bounds on their worst-case performance. We note their current intractability in high-dimensional settings and propose an approximate solution that is tractable, though the rate-distortion quality is impacted.

- We demonstrate the effectiveness by applying our method to lossy image compression. Concretely, we train a Probabilistic Ladder Network (Sønderby et al., 2016) on the CLIC (2018) dataset and show that the rate-distortion curve of our method is competitive with the state-of-the-art on the Kodak dataset (Eastman Kodak Company, 1999) on low bitrates.

## 2 LITERATURE REVIEW

### 2.1 TRANSFORM CODING

Neural network-based approaches lend themselves easiest to transform coding as a realisation of lossy compression. In transform coding (Goyal, 2001), given some data $\boldsymbol{x}$, it is first transformed using the *analysis transform* $\boldsymbol{z} = f_a(\boldsymbol{x})$ and then quantized $\hat{\boldsymbol{z}} = [\boldsymbol{z}]$, where $[\cdot]$ could denote rounding, or any other quantizer in general. The distribution of this quantized representation is modelled by some probability mass function $\tilde{P}(\hat{\boldsymbol{z}})$, which is used to encode $\hat{\boldsymbol{z}}$ using some entropy code. Note, that this is different from the true distribution of $\hat{\boldsymbol{z}}$, $P(\hat{\boldsymbol{z}})$, which is entirely determined by the distribution of $[f_a(\mathbf{x})]$, where $\mathbf{x} \sim p(\mathbf{x})$. During decompression, once $\hat{\boldsymbol{z}}$ is decoded, the *synthesis transform* $\boldsymbol{x}' = f_s(\boldsymbol{z})$ is applied to it to obtain a reconstruction of $\boldsymbol{x}$.

Given some metric $d(\cdot, \cdot)$, the *distortion* of the codec is

$$\mathbb{E}_{\mathbf{x} \sim p(\mathbf{x})}\left[d(\mathbf{x}, \mathbf{x}')\right], \tag{1}$$

and its rate is the expected codelength of the entropy code, which is

$$\mathbb{E}_{\mathbf{z} \sim P(\hat{\mathbf{z}})}\left[-\log \tilde{P}(\hat{\mathbf{z}})\right] = \mathbb{E}_{\mathbf{x} \sim p(\mathbf{x})}\left[-\log \tilde{P}(\hat{\mathbf{z}})\right]. \tag{2}$$

The equality holds, since $\hat{\mathbf{z}}$ is a deterministic transformation of $\mathbf{x}$.

### 2.2 BITS-BACK EFFICIENCY

If instead of mapping some data we wish to compress $\boldsymbol{x}$ to a single deterministic point $\boldsymbol{z}$ and entropy coding it by using some prior $p(\mathbf{z})$, we map it to an approximate posterior $q(\mathbf{z} \mid \boldsymbol{x})$, then we may select a random representative $\boldsymbol{z}^*$ from $q(\mathbf{z} \mid \boldsymbol{x})$ in the sense that $\boldsymbol{z}^*$ has a short Minimum Description Length (MDL, Grünwald et al. (2007)). Bits-back efficient methods are such that this MDL is approximately $\mathrm{KL}\left[\, q(\mathbf{z} \mid \boldsymbol{x}) \, || \, p(\mathbf{z}) \,\right]$. An important distinction should be made in whether we wish to communicate the distribution $q(\mathbf{z} \mid \boldsymbol{x})$ or a sample $\boldsymbol{z}^*$ drawn from it, as the communication problem

from which the MDL can be obtained, its precise value and the method required to obtain it are different.

The concept originates in the work of Hinton & Van Camp (1993). Given data $\mathbf{D}$ and model weights $\mathbf{w}$, they derive the MDL of the posterior weight distribution $q(\mathbf{w} \,|\, \mathbf{D})$ of a Bayesian Neural Network (BNN) with prior $p(\mathbf{w})$ to be $K = \mathrm{KL}\,[\,q(\mathbf{w} \,|\, \mathbf{D}) \,||\, p(\mathbf{w})\,]$. However, they did not give a method for achieving this, in fact, in their argument they merely show that in a longer message only $K$ nats are used to code the posterior.

In our setting, we are only interested in communicating a single sample $\boldsymbol{z}^*$ from $q(\mathbf{z} \,|\, \boldsymbol{x})$. We assume that we have access to a generative model $p(\mathbf{x} \,|\, \mathbf{z})p(\mathbf{z})$ and a posterior $q(\mathbf{z} \,|\, \mathbf{x})$. Then, the problem we must answer is what is the minimum expected number of nats $T[\mathbf{x} : \mathbf{z}]$ that is required to be communicated, such that we could draw a sample $\boldsymbol{z} \sim q(\mathbf{z} \,|\, \mathbf{x})$. This question was answered by Harsha et al. (2007), who proved the following theorem:

**Theorem 1** *(Harsha et al., 2007) Let the setting be as described above, and assume that the coder and the decoder share a common source of randomness (e.g. by using the same pseudo-random number generator with the same seed), then the minimum expected number of nats $T[\mathbf{x} : \mathbf{z}]$ that need to be communicated in order for the decoder to be able to sample a $\boldsymbol{z} \sim q(\mathbf{z} \,|\, \mathbf{x})$ can be bounded in terms of the mutual information $I[\mathbf{x} : \mathbf{z}]$ as*

$$I[\mathbf{x} : \mathbf{z}] \leq T[\mathbf{x} : \mathbf{z}] \leq I[\mathbf{x} : \mathbf{z}] + 2\log\left(I[\mathbf{x} : \mathbf{z}] + 1\right) + \mathcal{O}(1). \tag{3}$$

They prove the above claim constructively, by giving a rejection sampler that achieves this. This, to our knowledge, is the first example of what we call a *single-shot relative entropy coding* (REC) algorithm, and we will examine it in further detail in Section 4. Bits-back efficiency comes from the fact that $I[\mathbf{x} : \mathbf{z}] = \mathbb{E}_{\mathbf{x} \sim p(\mathbf{x})}\,[\mathrm{KL}\,[\,q(\mathbf{z} \,|\, \mathbf{x}) \,||\, p(\mathbf{z})\,]]$.

Relative entropy coding subsumes entropy coding in the sense that any entropy code is a relative entropy code as well, since for a discrete $\hat{\mathbf{z}} \sim P(\hat{\mathbf{z}})$ and the point-mass $\delta_{\hat{\mathbf{z}}}$ on $\hat{\mathbf{z}}$,

$$-\log P(\hat{\boldsymbol{z}}) = \mathrm{KL}\,[\,\delta_{\hat{\mathbf{z}}} \,||\, P(\hat{\boldsymbol{z}})\,]. \tag{4}$$

See Havasi et al. (2018) for details on the derivation. In this sense, methods that use quantization and then entropy coding are also bits-back efficient, but are restricted to point-masses for their latent posteriors. Thus, one of the advantages of our method lies in that it allows a much wider class of posteriors to be used.

## 2.3 RELATED WORKS

Minimal Random Code Learning (Havasi et al., 2018) is a powerful coding technique that allows us to code a random sample from a continuous distribution, rather than coding a deterministic sample, which requires quantization, as any fixed sample would have 0 density. They use their technique to compress the weights of BNNs. The structure of the weight space is much different from the structure of the latent space of a generative model trained for the compression of some medium. Hence, we argue that their practical compression procedure is infeasible for our setting, the random fixed-size block partitioning as well as the retraining between the coding of blocks is incompatible with our setting as well as computationally infeasible.

Townsend et al. (2019) were the first to develop a bits-back efficient image compression algorithm using VAEs, called Bits-back with ANS (BB-ANS). However, they develop a lossless compression algorithm, and hence the performance their approach is not comparable to ours. The sense in which their method is "bits-back" also differs from ours. Townsend et al. (2019) stick to the original bits-back setting. Concretely, they show a clever way of utilizing the overhead of the method of Hinton & Van Camp (1993) for efficient coding, our work on the other hand, avoids sending it in the first place. Their coding efficiency collapses onto entropy efficiency when coding a single image and bits-back efficiency is only achieved when coding larger batches.

Lately, there has been an explosion of interest in using powerful deep generative models for lossy image compression, and thus all previous methods had to deal with the issue of the non-differentiability of quantization. These approaches can generally be put into three different categories, depending on how they avoid this problem. There are approaches that waive the paradigm of end-to-end training, e.g. by not incorporating the feedback from the gradient of the rate term into the optimization of the

entropy coder, either by using inflexible distributions without any parameters (Toderici et al., 2017), or by optimizing them post-training (Johnston et al., 2018; Rippel & Bourdev, 2017). The issue with this, as pointed out in Ballé et al. (2018) is that not training end-to-end might severely limit the efficiency of these methods.

Another approach of getting aroud this issue has been to learn to generate samples from the image distribution $p(\mathbf{x})$, using a Generative Adversarial Network (GAN, Goodfellow et al. (2014)), by learning the synthesis transform first and using an auxiliary discriminator. Then, fixing the discriminator, the analysis transform is trained in order to minimize the reconstruction error $d(\boldsymbol{x}, f_s(f_a(\boldsymbol{x})))$. The noise provided $f_s$ during the initial GAN training should be a continous relaxation of quantization noise, such as a uniform distribution (Ballé et al., 2016).

Finally, there are methods that stick to Variational Auto-Encoders (VAEs, Kingma & Welling (2014); Higgins et al. (2017)), optimize the weighted rate-distortion directly, and replace the non-differentiable quantities with smooth approximations during training. Theis et al. (2017) introduce *compressive auto-encoders*, and replace the derivative of the quantization operation with the identity function, and the rate of the discrete $\hat{\mathbf{z}}$ by a differentiable upper bound which is used in the loss. Closest to our work are the works of Ballé et al. (2016; 2018); Minnen et al. (2018), who all model the quantization noise as uniform noise centered on the quantized representation. Ballé et al. (2016) are the first to derive the connection between the weighted rate-distortion loss and the $\beta$-Evidence Lower Bound ($\beta$-ELBO, Higgins et al. (2017)), though they only derive it for a smaller class of distributions than we do, as well as through a different route, as we arrive at it through the MDL principle (Grünwald et al., 2007). Crucially, these methods are still only equivalent to VAEs during training, and the continuous relaxations are switched back to the original discrete operations thereafter.

## 3    COMPRESSION WITHOUT QUANTIZATION

Our method relies on two observations. First, we note that the sole purpose of quantization is that once we have obtained a deterministic representation $\hat{\boldsymbol{z}}$, it has non-zero probability mass, and hence entropy coding may be used to compress it. Hence, if we could replace entropy coding with a method that could use a probability density $p(\mathbf{z})$ instead of the mass $\tilde{P}(\hat{\mathbf{z}})$, then we could forgo the quantization step and thus our training objective becomes end-to-end differentiable.

Second, as we have seen in Section 2.2, if in addition have access to an approximate latent posterior $q(\mathbf{z} \,|\, \mathbf{x})$, then it is enough to communicate approximately

$$I[\mathbf{x} : \mathbf{z}] = h[\mathbf{z}] - h[\mathbf{z} \,|\, \mathbf{x}] = \mathbb{E}_{\boldsymbol{x} \sim p(\mathbf{x})} \left[ \text{KL} \left[ q(\mathbf{z} \,|\, \mathbf{x}) \,||\, p(\mathbf{z}) \right] \right] \tag{5}$$

nats such that the decoder may obtain a sample $\boldsymbol{z} \sim q(\mathbf{z} \,|\, \boldsymbol{x})$. Applying $f_s$ to $\boldsymbol{z}$ could then be used to reconstruct $\boldsymbol{x}$.

These two could be combined by noting that our setting fulfils all premises required for both observations to hold. This enables the following development of a lossy transform codec:

**Training**    After fixing an appropriate architecture for $f_a(\boldsymbol{x}) = q(\mathbf{z} \,|\, \boldsymbol{x})$ and $f_s$ such that the likelihood is $p(\mathbf{x} = f_s(\boldsymbol{z}) \,|\, \boldsymbol{z})$, we train our model using the $\beta$-Evidence Lower Bound (Higgins et al., 2017):

$$\mathbb{E}_{\mathbf{x}, \mathbf{z} \sim q(\mathbf{z} \,|\, \mathbf{x})p(\mathbf{x})} \left[ -\log p(\mathbf{x} \,|\, \mathbf{z}) \right] + \beta \mathbb{E}_{\mathbf{x} \sim p(\mathbf{x})} \left[ \text{KL} \left[ q(\mathbf{z} \,|\, \mathbf{x}) \,||\, p(\mathbf{z}) \right] \right]. \tag{6}$$

Note, that if the distortion metric $d(\cdot, \cdot)$ is such, that $\exp\{-d(\boldsymbol{x}, \boldsymbol{x}')\}$ can be normalized, then the negative log-likelihood can be identified with the distortion $D$. $\mathbb{E}_{\mathbf{x} \sim p(\mathbf{x})} \left[ \text{KL} \left[ q(\mathbf{z} \,|\, \mathbf{x}) \,||\, p(\mathbf{z}) \right] \right] = I[\mathbf{x} : \mathbf{z}]$ can also be identified with the rate $R$. Hence, in this case the negative $\beta$-ELBO is equivalent to training for the rate-distortion loss

$$L = D + \beta R, \tag{7}$$

which was our aim to train for in the first place. This gives us a new way to interpret $\beta$ in Equation 6: it allows the model to converge on different points on its rate-distortion curve (see Ballé et al. (2016) for the full argument).

Note, that even if normalizing $\exp\{-d(\boldsymbol{x}, \boldsymbol{x}')\}$ is intractable $\beta$ can be adjusted in Equation 6 to compensate for this unknown constant during training.

**Compression and Decompression**  For given $\boldsymbol{x}$ that we wish to compress, we calculate $q(\mathbf{z} \mid \boldsymbol{x})$ using the analysis transform. Then, we use relative entropy coding (REC) to code a sample $\boldsymbol{z} \sim q(\mathbf{z} \mid \boldsymbol{x})$. The decoder can then re-obtain $\boldsymbol{z}$ and apply the synthesis transform to it to reconstruct $\boldsymbol{x}$.

## 4 RELATIVE ENTROPY CODING

As mentioned in the previous section, our method relies on the existence of relative entropy coding (REC) in order to be able to code the latent representation of a given input $\boldsymbol{x}$ with bits-back efficiency. To our knowledge, the first such algorithm was developed by Harsha et al. (2007), as a variant of rejection sampling. This algorithm is unfortunately intractable in more than a few dimensions, as it requires to keep track of probabilities across the entire range of $\hat{\mathbf{z}}$.

The simplest way to extend the univariate algorithm to the multivariate setting is to note the independence assumption for the dimensions of $\hat{\mathbf{z}}$ and thus concatenating univariate samples for each dimension will given an exact multivariate sample. Hence, we could use REC to encode the dimensions of $\hat{\mathbf{z}}$ individually, i.e. for each $\hat{z}_i$ obtain some code $c_i$, and build an empirical distribution $P(c)$ over these. Then, we can encode each $c_i$ using entropy coding. The issue with this approach is that while the upper bound in Theorem 1 is quite tight if we apply it to the whole multivariate distribution over $\hat{\mathbf{z}}$, when applied to each of its dimensions $\hat{z}_i$, the $\mathcal{O}(1)$ cost is incurred for each dimension, which makes this approach very inefficient. Hence, we focus on multivariate REC.

### 4.1 ADAPTIVELY GROUPED IMPORTANCE SAMPLING

---

**Algorithm 1** Importance sampling algorithm proposed by Havasi et al. (2018)

**Inputs:**
$P$ - Proposal distribution
$Q$ - Target Distribution
$\langle x_i \sim Q \mid i \in \mathbb{N} \rangle$ - Shared sequence of random draws from $Q$

**procedure** IMPORTANCE-SAMPLER($P, Q, \langle x_i \sim Q \mid i \in \mathbb{N} \rangle$)
    $K \leftarrow \exp\{\mathrm{KL}\,[\,Q \,\|\, P\,]\}$
    $\tilde{w}_i \leftarrow \log \frac{Q(x_i)}{P(x_i)} \quad \forall i = 1, \ldots K$
    $j \leftarrow \arg\max_i \tilde{w}_i$
    **return** $j, x_j$
**end procedure**

---

The basis of our approach is the approximate importance sampler proposed by Havasi et al. (2018). For given $\boldsymbol{x}$, it proceeds by letting $K = \mathrm{KL}\,[\,q(\mathbf{z} \mid \boldsymbol{x}) \,\|\, p(\mathbf{z})\,]$ and drawing $S = \lceil e^K \rceil$ samples $(\boldsymbol{z}_1, \boldsymbol{z}_2, \ldots, \boldsymbol{z}_S)$ from $p(\mathbf{z})$. Then the index of the maximal importance weight

$$c^* = \underset{c \in \{1,2,\ldots S\}}{\arg\max} \frac{q(\boldsymbol{z}_c \mid \boldsymbol{x})}{p(\boldsymbol{z}_c)} \tag{8}$$

is used as the encoding of $\boldsymbol{x}$. The decoder can recover $\boldsymbol{z}_{c^*}$ by drawing $c^*$ samples from $p(\boldsymbol{z})$ and then can use it to reconstruct $\boldsymbol{x}$, assuming it can generate the same sequence of samples. Clearly, by virtue of the choice for $S$, $c^*$ can be coded in $\mathrm{KL}\,[\,q(\boldsymbol{z} \mid \boldsymbol{x}) \,\|\, p(\boldsymbol{z})\,]$ nats, and Havasi et al. (2018) also show that $\boldsymbol{z}_{c^*}$ has negligible bias. The algorithm is depicted in Algorithm 1, note that Havasi et al. (2018) originally sample the softmax distribution of the $\tilde{w}_i$. We found it better to select samples biased towards the mode by always selecting the most likely sample. Though now we do not need to keep track of the probabilities over the entire range of $\mathbf{z}$, the run-time is still going to be exponential in $K$, and hence becomes prohibitive for $K$ larger than 15 - 20. In our experiments we had to deal with KLs well above $10^5$, and hence solving this issue is critical for the viability of our method.

To solve this issue, Havasi et al. (2018) randomly partition the vector of interest $\boldsymbol{w}$ into fixed size blocks, in the hope that on average each individual block will have approximately some $C$ nats of total KL. They then code each the blocks sequentially. The concern is that the next block to be coded is a suboptimal choice, i.e. its total KL is significantly higher or lower than the desired $C$ nats. To circumvent this problem, they introduce a blockwise KL penalty to the loss that forces each block

to have equal KL, and they intersperse coding a block with several rounds of retraining. They show that this solves the suboptimal choice problem, and their coding procedure also does not impact the quality of the coded $\boldsymbol{w}$.

Unfortunately, this fix is not applicable in our case. Havasi et al. (2018) can use the retraining trick to compensate for suboptimal block choices. However, if we hope to obtain a useful compression codec, we should avoid extremely expensive operations to code a single item, such as performing gradient descent for each individual input $\boldsymbol{x}$ to our algorithm.

To remedy these issues, instead of selecting fixed-size blocks randomly we impose a group size constraint of $G$ bits and total group KL constraint of $B$ bits. Then, we partition the dimensions of $\mathbf{z}$ such that each set in the partition (what we call a group) has size at most $2^G$ and each group has total KL at most $K$ bits. The details are depicted in Algorithm 2.

---

**Algorithm 2** Adaptively grouped importance sampler

**Inputs:**
$K$ - Maximum individual KL allowed
$G$ - Maximum group size
$B$ - Bit budget per group
$P$ - Proposal distribution
$Q$ - Target Distribution
$\langle x_i \sim Q \,|\, i \in \mathbb{N} \rangle$ - Shared sequence of random draws from $Q$

**procedure** ADAPTIVE-IMPORTANCE-SAMPLER($K, G, B, P, Q, \langle x_i \sim Q \,|\, i \in \mathbb{N} \rangle$)
   $\Gamma \leftarrow ()$         ▷ Initialize empty list of group sizes
   $k_i \leftarrow \mathrm{KL}\,[\,Q_i \,||\, P_i\,]\,\forall i = 1, \ldots N$     ▷ Get KLs for each dimension
   $Q', P', O \leftarrow$ REMOVE-OUTLIERS($\{k_i\}_{i=1}^N, K$)
   $\gamma \leftarrow 0, k \leftarrow 0$        ▷ Current group size and group KL
   **for** $i \leftarrow 1, \ldots \dim(Q')$ **do**
      **if** $k + kl_i > B$ or $\gamma + 1 > G$ **then**
         Append $\gamma$ to $\Gamma$
         $k \leftarrow kl_i, \gamma \leftarrow 1$
      **else**
         $k \leftarrow k + kl_i, \gamma \leftarrow \gamma + 1$
      **end if**
   **end for**
   Append $\gamma$ to $\Gamma$        ▷ Append the last group size

   $S = (), I = (), g \leftarrow 0$     ▷ Samples, sample codes and current group index
   **for** $\gamma$ in $\Gamma$ **do**       ▷ Now importance sample each group
      $i, s \leftarrow$ IMPORTANCE-SAMPLER($P_{g:g+\gamma}, Q_{g:g+\gamma}, \langle x_i \sim Q \,|\, i \in \mathbb{N} \rangle$)
      Append $i$ to $I$, append $s$ to $S$
      $g \leftarrow g + \gamma$
   **end for**
   **return** I, S, O
**end procedure**

---

While the concern of intractability due to high KL is resolved this way but we are faced with two inefficiencies. First, for groups whose size is much smaller than $2^G$, dedicating $G$ bits to coding the group size is wasteful. Second, many groups will have size $2^G$ but total KL much less then $K$, and hence dedicating $K$ bits to code the sample is wasteful. To solve these issues, we built an empirical distribution over group sizes using our training dataset, which is tractable since we chose $G$ to be not too large ($2 \le G \le 6$), and then used arithmetic coding (Rissanen & Langdon, 1981) to code the group sizes. This worked very effectively, to the extent that for appropriately selected $G$ the cost of communicating the group sizes is effectively negligible.

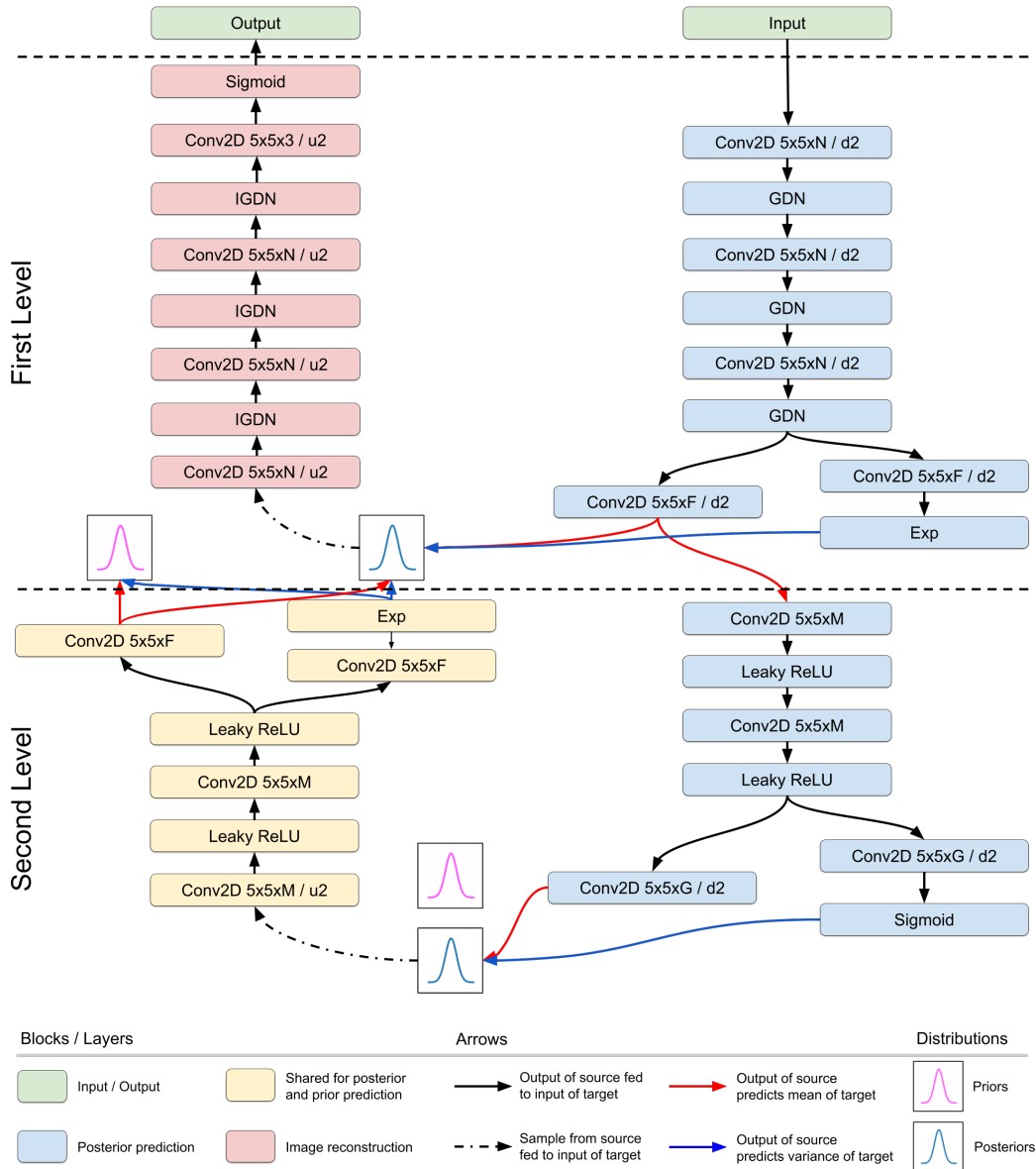

Figure 1: PLN network architecture. The blocks signal data transformations, the arrows signal the flow of information. **Block descriptions:** *Conv2D:* 2D convolutions along the spatial dimensions, where the $W \times H \times C/S$ implies a $W \times H$ convolution kernel, with $C$ target channels and $S$ gives the downsampling rate (given a preceding letter "d") or the upsampling rate (given a preceding letter "u"). If the slash is missing, it means that there is no up/downsampling. All convolutions operate in `same` mode with zero-padding. *GDN / IGDN:* these are the non-linearities described in Ballé et al. (2016). *Leaky ReLU:* elementwise non-linearity defined as $\max\{x, \alpha x\}$, where we set $\alpha = 0.2$. *Sigmoid:* Elementwise non-linearity defined as $\frac{1}{1+\exp\{-x\}}$. We ran all experiments presented here with $N = 196, M = 128, F = 128, G = 24$.

The relative entropy codes of the groups were also further compressed using Elias $\delta$-coding (Elias, 1975). This is the main point of inefficiency of this method, as the empirical distribution of sample indices has a much heavier tail than the distribution implied by $\delta$-coding. Concretely, we found that using $\delta$-coding the resulting code lengths are approximately $1.6 - 1.8$ times the theoretically possible REC code lengths. However, building an empirical distribution over the sample indices and using it for entropy coding is more challenging, as the number of possible indices for a "useful" $K$ is between $2^{15} \leq 2^K \leq 2^{20}$. This means that to obtain a good estimate of the tail mass, i.e. of indices above $2^{14} - 2^{15}$, we need a lot of data, or some reasonable smoothing should be used, and we leave this for future work.

## 5 EXPERIMENTS

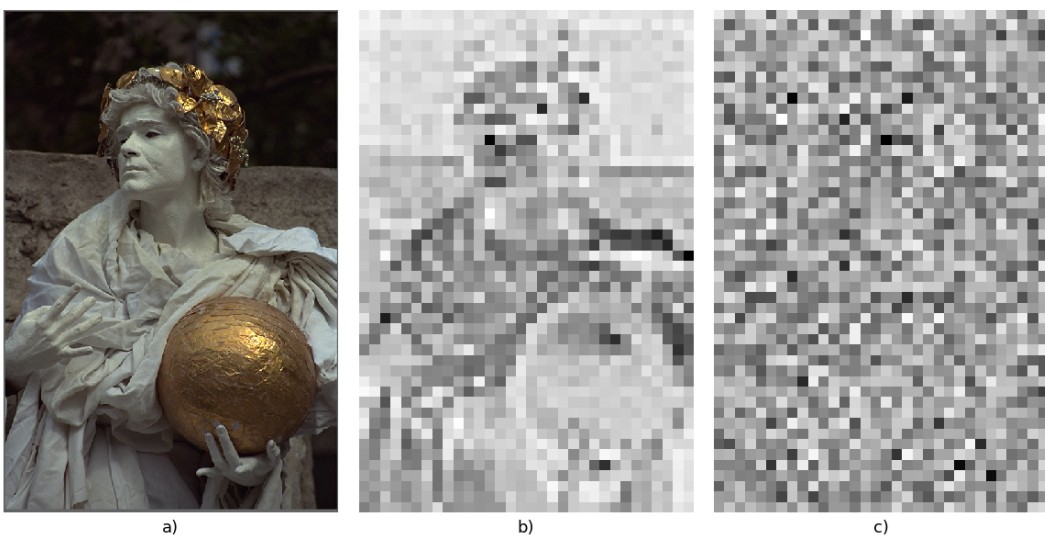

Figure 2: Demonstration of the effectiveness of the PLN's conditional independence assumption $p(\mathbf{z}) = p(\mathbf{z}^{(1)} \,|\, \mathbf{z}^{(2)}) p(\mathbf{z}^{(2)})$. **a)** Original image from the Kodak dataset. **b)** Randomly selected channel of a latent sample $\boldsymbol{z}^{(1)} \sim q(\mathbf{z}^{(1)} \,|\, \boldsymbol{z}^{(2)}, \boldsymbol{x})$, clearly there is a lot of structure present in the sample. **c)** The same channel of the latent sample standardized according to $p(\mathbf{z}^{(1)} \,|\, \boldsymbol{z}^{(2)})$, i.e. we $\frac{z_i^{(1)} - \mu_i}{\sigma_i}$ is displayed for each dimension, where the $\mu_i$ and $\sigma_i$ are the means and standard deviations of the dimensions of $p(\mathbf{z}^{(1)} \,|\, \boldsymbol{z}^{(2)})$, respectively. We see that relative to the conditional prior, the latent structure is effectively captured.

We applied our framework to lossy image compression. We followed the work of Ballé et al. (2016) and Ballé et al. (2018). Not only do their results represent the current state-of-the-art, but their work is also the closest to ours. In particular, as mentioned before, to our knowledge they were the first to derive the connection of the weighted rate-distortion loss with the $\beta$-ELBO as the training objective for a VAE. However, we stress that they derived this for a VAE whose latent distribution was a continuous relaxation of the quantization error. Therefore, their results were restricted to a narrow set of distributions, and neither did they make the connection to bits-back efficientcy. Most notably though, they only used the continuous relaxation during the training of the model, thereafter switching back to quantization and entropy coding, which, they show does not impact the predicted performance, and hence confirming that their relaxation during training is reasonable. This is in striking contrast with our method, in which not only do we allow any latent distributions for which the KL can be calculated, but our architecture also remains the same after training.

Nonetheless, we based our architecture entirely on the one they present in Ballé et al. (2018), appropriately modified for our setting. Most importantly, where they use i.i.d. uniform distributions for the posterior and complicated non-parametric priors, we opted for Gaussians, a much more standard

choice for VAEs. We also opted for a similar hierarchical VAE structure, which in our case is a Probabilistic Ladder Network (PLN, Sønderby et al. (2016)). The reason for this is as shown by Ballé et al. (2018) and also confirmed by us empirically, the full independence assumption in the latent distribution of a VAE is very limiting, as there is a lot of topological structure even on the latent level. One solution to introduce a rich dependence structure is to introduce further latent variables, conditioned upon which the rest are assumed to be independent (Bishop, 1998). Concretely, we break up our generative model $p(\mathbf{x}, \mathbf{z}) = p(\mathbf{x} \mid \mathbf{z}^{(1)})p(\mathbf{z}^{(1)} \mid \mathbf{z}^{(2)})p(\mathbf{z}^{(2)})$, and our approximate latent posteriors now become $q(\mathbf{z}^{(2)} \mid \mathbf{x})$ and $q(\mathbf{z}^{(1)} \mid \mathbf{x}, \mathbf{z}^{(2)}) \propto q(\mathbf{z}^{(1)} \mid \mathbf{x})p(\mathbf{z}^{(1)} \mid \mathbf{z}^{(2)})$. Further, our architecture also only uses convolutions and deconvolutions as non-linearities. This allows us to compress arbitrary sized images, and the latent space can naturally grow with the dimensions of the images. Full details of the architecture can be seen in Figure 1. A confirmation that the PLN's conditional independence structure efficiently removes the topological structure in the latent space on the first stochastic level is shown in Figure 2. Note, that in order for us to code a latent sample $\boldsymbol{z} \sim q(\mathbf{z} \mid \boldsymbol{x})$, we first code $\boldsymbol{z}^{(2)} \sim q(\mathbf{z}^{(2)} \mid \boldsymbol{x})$ using $p(\mathbf{z}^{(2)})$ and then code $\boldsymbol{z}^{(1)} \mid \boldsymbol{z}^{(2)} \sim q(\mathbf{z}^{(1)} \mid \boldsymbol{x}, \boldsymbol{z}^{(2)})$.

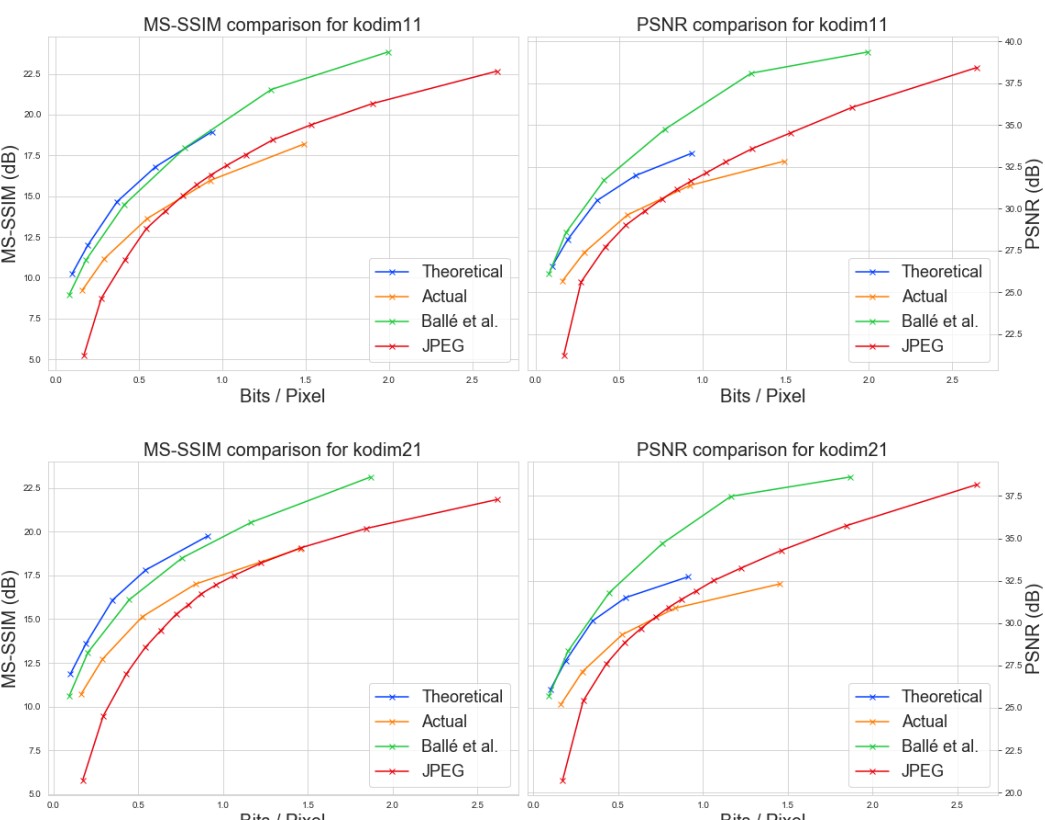

Figure 3: Comparison of rate-distortion curves on `kodim11.png` and `kodim21.png` from the Kodak dataset. MS-SSIM results are presented in decibels, where the conversion is done using the formula $-10 \cdot \log_{10} \left(1 - \text{MS-SSIM}(\boldsymbol{x}, \boldsymbol{x}')\right)$ as is done by Ballé et al. (2018). For the description of the labels, see Section 5.

In our experiments, we compare the performance of our method with the method of Ballé et al. (2018) and JPEG as a baseline on their rate-distortion curves, using the Peak Signal-to-Noise Ratio (PSNR) and Multiscale Structural Similarity Index (MS-SSIM) perceptual metrics (Huynh-Thu & Ghanbari, 2008; Wang et al., 2003) as distortion distances. Our models were trained using $\beta \in \{300, 600, 1200, 2400, 4800\}$ and the Mean Absolute Error (MAE) loss. The reason for the MAE loss is due to Zhao et al. (2015), who show that using it greatly improves image reconstruction quality at no significant extra cost. The model of Ballé et al. (2018) was trained using $\lambda \in \{0.001, 0.003, 0.01, 0.03, 0.1, 0.3\}$ and the Mean Squared Error (MSE) loss. Both were trained on the CLIC (2018) training dataset, which allows a better comparison in performance. All models

were trained for $2 \times 10^5$ iterations with Adam (Kingma & Ba, 2014), using a learning rate of $10^{-4}$. We stress, that for comparability, the architectures were made completely identical, except for the latent distributions, which needed adaptation to our setting.

We report the rate-distortion curves on two images taken from the Kodak dataset in Figure 3. Further comparison plots can be found in Appendix A. For our models, we report the theoretically optimal rate-distortion, the rate of which for an image $x$ can be calculated as $\mathrm{KL}[\,q(\mathbf{z}\,|\,\mathbf{x}=x)\,\|\,p(\mathbf{z})\,]/\log 2$, and the actual rate-distortion achieved by our proposed REC algorithm from Section 4.1. For the models of Ballé et al. (2018), we report the actual rate-distortion.

We see that our method is competitive with the method of Ballé et al. (2018) on low bitrates and starts tailing off more on higher bitrates.

## 6 CONCLUSION

We develop a lossy transform coding framework based on generative models equipped with an approximate latent posterior and show that it is bits-back efficient (Hinton & Van Camp, 1993) and can also be optimized end-to-end. We propose a relative entropy coding algorithm to achieves bits-back efficiency, and we demonstrate the efficiency of our method by training a Probabilistic Ladder Network on the CLIC (2018) dataset and show that it is competitive with the current state-of-the-art on the Kodak dataset Eastman Kodak Company (1999).

While allowing for a much wider class of latent distributions, our methods relies on the existence of efficient relative entropy coding algorithms for the selected distributions. Our REC algorithm is based on the importance sampler proposed by Havasi et al. (2018), and while it is tractable and works reasonably well on low bitrates, its rates are approximately $1.6 - 1.8$ times higher than the theoretically possible ones. Currently the run-time is also slower than other methods, it takes around 1-5 minutes to compress a reasonably sized image. Improving the rate factor and the run-time does not seem too difficult a task, but since the focus of our work was to demonstrate the efficiency of relative entropy coding, it is left for future work.

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

## A FURTHER COMPARISONS

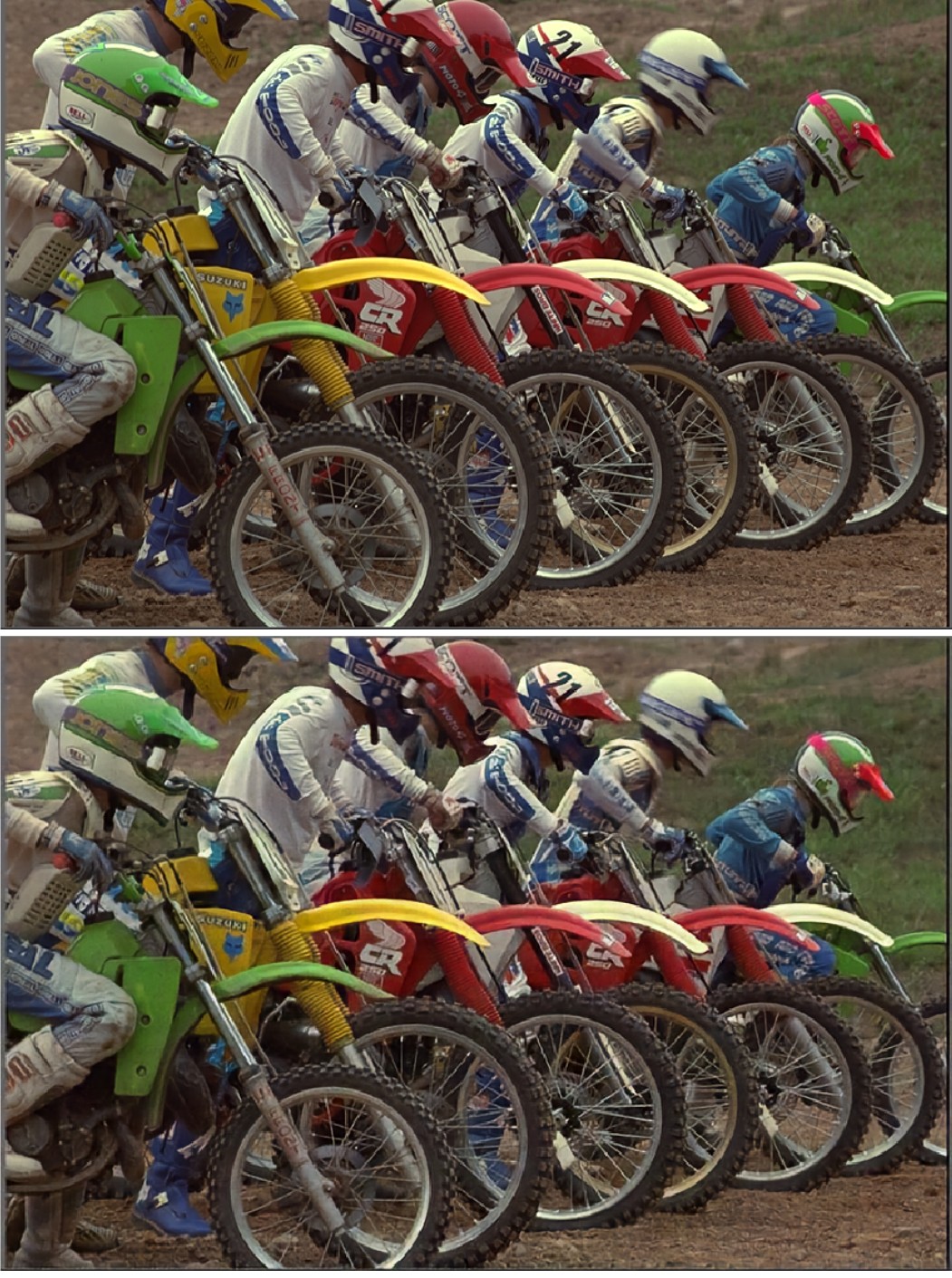

Figure 4: Reconstruction comparison for `kodim05.png`. Top: original image. Bottom: Reconstruction using our PLN with $\beta = 600$.

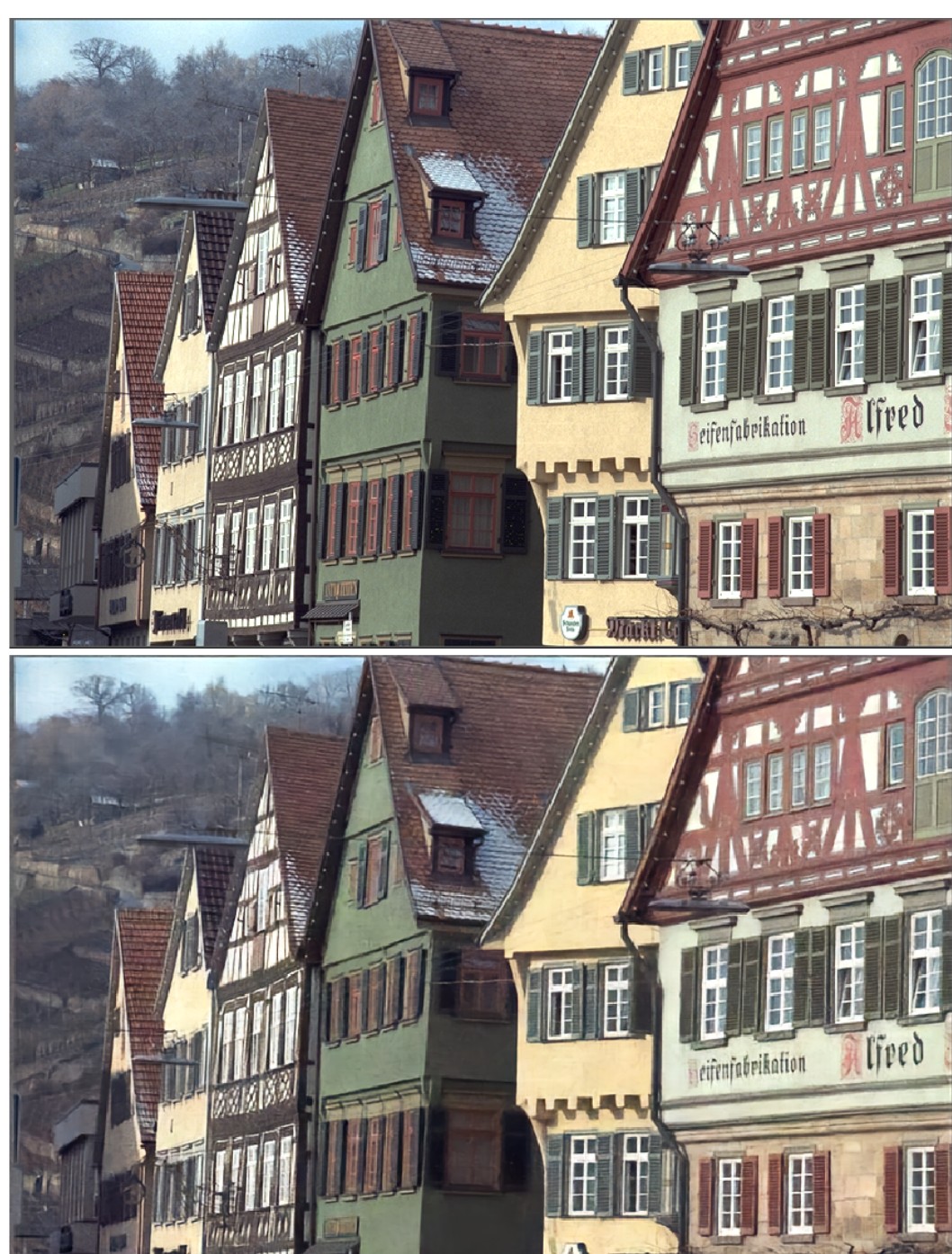

Figure 5: Reconstruction comparison for kodim08.png. Top: original image. Bottom: Reconstruction using our PLN with $\beta = 1200$.

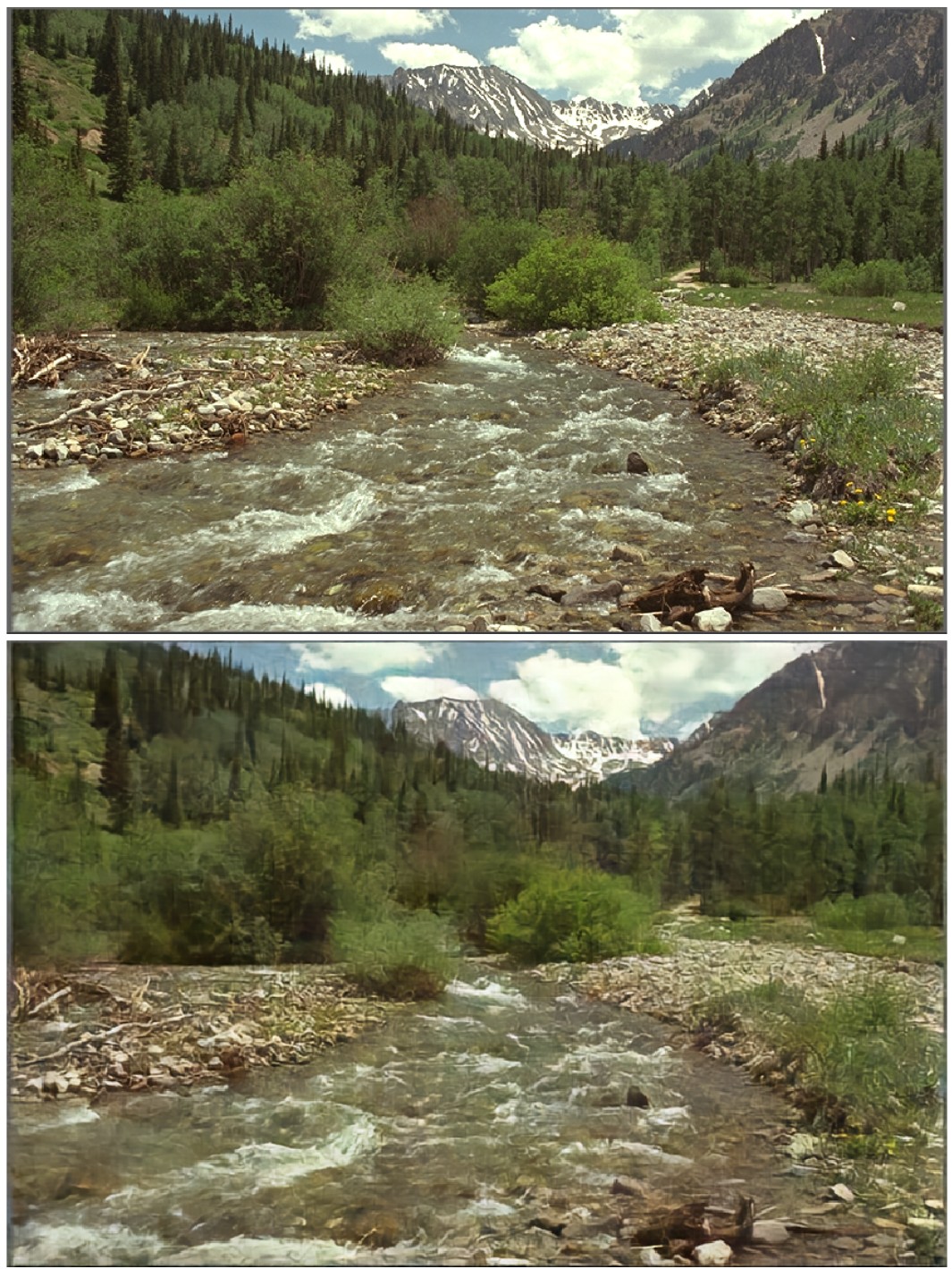

Figure 6: Reconstruction comparison for kodim13.png. Top: original image. Bottom: Reconstruction using our PLN with $\beta = 2400$.

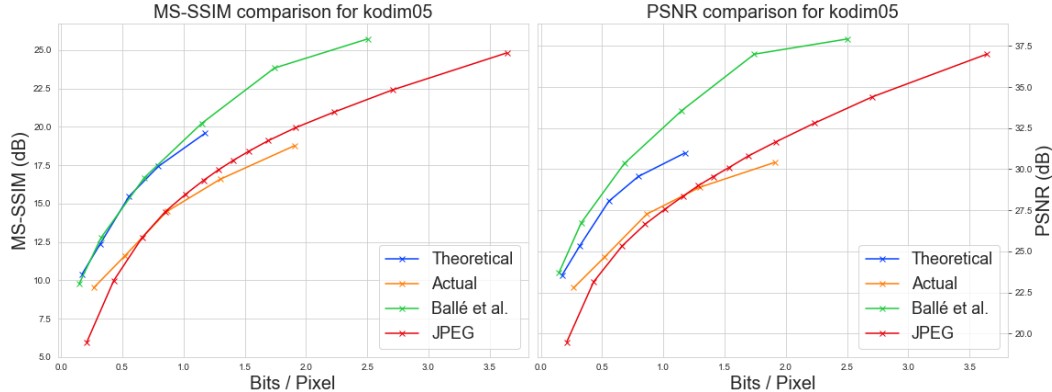

Figure 7: Rate-distortion curves for `kodim05`

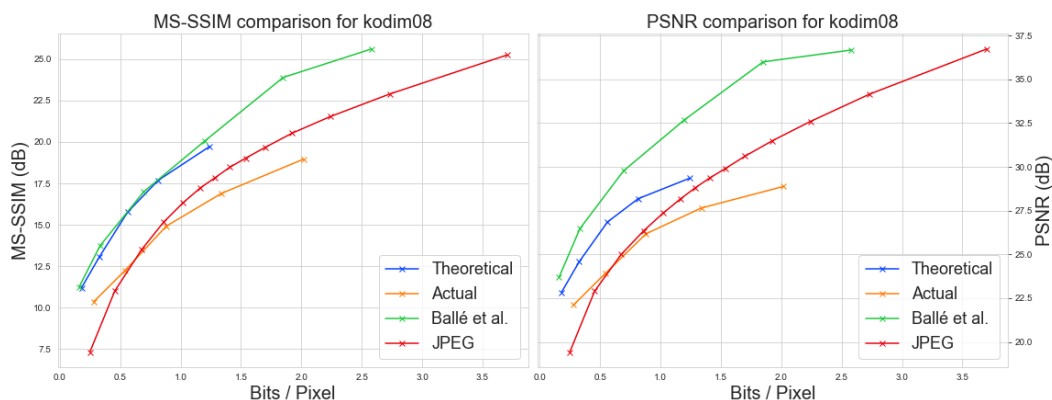

Figure 8: Rate-distortion curves for `kodim08`

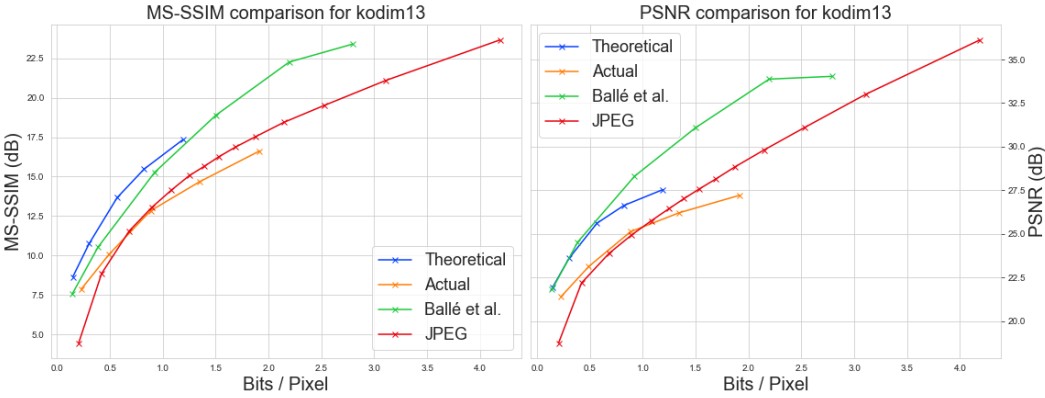

Figure 9: Rate-distortion curves for `kodim13`

