# OpenReview forum: "Compression without Quantization"
_ICLR.cc/2020/Conference — Reject_

### Official Review · AnonReviewer3 · 2019-10-23
**Official Blind Review #3**

**Rating:** 3

**Review:**

Summary:
This paper aims to circumvent the quantization step and associated gradient approximations for compression algorithms that make use of entropy coding for compression. Entropy coding requires a probability mass function over discrete symbols. As an alternative approach the authors adapt the MIRACLE algorithm by Havasi et al. (2018), which was originally used to compress Bayesian neural networks, to work for lossy source compression with probability density functions over continuous latent variables.
The algorithm is based on taking several importance samples from the prior p(z) with the encoding distribution q(z|x) as the target distribution. The number of samples is equal to the exponent of the KL between the encoding and prior distribution, which can quickly grow to an uncontrollably large number. The index of the sample with the maximum importance weight is used as the code for the original image. If both sender and receiver have access to the same random sampler, the receiver can reproduce the sample by drawing a number of samples equal to this index, and decoding the last sample. Similar in spirit to Havasi et al, the authors take several steps to ensure that the KL divergence (and therefore the number of samples) does not become prohibitively large. The compression performance is evaluated by training the proposed model and the competing neural network-based method [1] on the Clic dataset, and evaluating it on a subset of the images of the Kodak dataset. JPEG is also used as a baseline. The authors show results achievable if coding using the relative entropy was perfect (denoted with ‘theoretical’), and the practically achieved compression performance (‘actual’).

Decision:
Weak reject: although the idea of circumventing the quantization step required by the use of entropy coders is certainly valid and interesting, the results in the paper show that the resulting compression performance is worse than the competing method that does quantize. Moreover, the encoding time is significantly longer than this same baseline.

Supporting arguments for decision:
Although the motivation for circumventing the quantization step seems plausible, the authors show no evidence that the competing method [1], which does perform a post-training quantization step, actually suffers from it. The authors even state on page 8 that “ Most notably though, they only used the continuous relaxation during the training of the model, thereafter switching back to quantization and entropy coding, which, they show does not impact the predicted performance, and hence confirming that their relaxation during training is reasonable.“ If post-training quantization is reasonable, then this overthrows the entire motivation. More importantly, the “theoretically” achievable results of the proposed method in seem only competitive in the low-bit rate regime and worse in the higher bit rate regimes. Even more, the “actual” practically achieved compression results are worse than [1] and also considerably worse than the “theoretically” achievable compression results. The authors do provide a reason for why the “theoretical” and “actual” results are so far apart, but are unfortunately not able to overcome this issue.
In the conclusion the authors honestly admit that the runtime of their method is much slower than the competitors (1-5 min for proposed method vs ~0.5 s  for [1] for encoding times). I appreciate that the authors mention this. The authors then state “improving the rate factor and the run-time does not seem too difficult a task, but since the focus of our work was to demonstrate the efficiency of relative entropy coding, it is left for future work.” I do not think this paper demonstrates the efficiency of relative entropy coding, the results simply don’t support this claim, and I therefore think that stating that the issues seem not too difficult to overcome is insufficiently convincing.

The quality of the empirical study can be improved. Another neural network-based compression baseline would make the empirical evaluation of the proposed method more insightful. Now we only see that the result is worse than [1], but it would be good to know how it compares to other baselines such as [2]. Furthermore, the paper does not show compression results aggregated over the entire Kodak dataset, but rather picks 2 images for the main part of the paper, and shows 3 in the appendix. Showing aggregate results gives a more robust estimate of the performance. Individual image results can just be put in the appendix.


Additional feedback to improve paper (not part of decision assessment):
- In section 5 on the dependency structure of latents in the ladder VAE: is it really necessary to indicate the dependency structure with “topological structure”? Seems unnecessary to me as dependency structure is a clear enough description already without making it sound overly complicated.
- Page 9: “Further, our architecture also only uses convolutions and deconvolutions as non-linearities.” Convolutions and deconvolutions are not non-linearities.
- Fig 2: I’m not sure if this figure is relevant enough for such a prominent placement in the paper. It doesn’t discuss anything relevant to the contributions claimed in this paper.
- I can’t find a definition of O in line 3 of “procedure” in algorithm 2 and in the return statement.


[1] Johannes Ballé, David Minnen, Saurabh Singh, Sung Jin Hwang, and Nick Johnston. Variational image compression with a scale hyperprior. ICLR 2018.
[2] Lucas Theis et al.  Lossy Image Compression with Compressive Autoencoders. ICLR 2017


**Experience Assessment:**

I have published one or two papers in this area.

**Review Assessment: Checking Correctness Of Derivations And Theory:**

I assessed the sensibility of the derivations and theory.

**Review Assessment: Checking Correctness Of Experiments:**

I carefully checked the experiments.

**Review Assessment: Thoroughness In Paper Reading:**

I read the paper at least twice and used my best judgement in assessing the paper.

---

> ### Author Response · Authors · 2019-11-12
> **Responses to Reviewer #3**
>
> We thank the reviewer for their detailed comments on our work.
>
> We agree with the reviewer that in our paper we show no evidence that competing methods suffer from quantization, in particular, we do not believe that they suffer from it.
>
> Our work is simply focussed on an alternative, novel approach to lossy compression, that allows a much wider class of algorithms to be used as the encoder and decoder. Concretely, previous approaches required specific assumptions about the compression pipeline (e.g. what kind of quantization is performed. In [2] it is assumed to be rounding, and hence its derivative is replaced by a continuous relaxation that the authors had to choose) or the model (e.g. [1] relies on the latent posteriors to be uniform distributions such that their terms cancel in the ELBO). In contrast, our method works for any generative model where an approximate posterior for the latents is available. This means that an arbitrary valid VAE could be used in our method, with complete freedom of choice for both the latent posteriors and priors.
>
> We, therefore, believe that the removal of restrictions on the latent space's distributions is a strong motivation, and by using a more flexible family for our VAE, the results of [1] could be surpassed.
>
> Indeed, the model performance degrades more at higher rates than the performance of [1]. A simple explanation might be that of model capacity: our model has been trained with much fewer latent filters than what [1] used (e.g. we used 24 latent filters as opposed to the 128 and 196 that Balle used for lower and higher rates, respectively).
>
> We agree that a better empirical study could be performed. The main difficulty we found was assessing how different training sets might impact the model performance, and hence decided that the fairest comparison would be if all models were trained on the same dataset. Sadly, this limits the comparisons to works where the code to achieve the reported results is freely available, which was only true in the case of [1].
>
> We opted to report single image statistics only as we believe that aggregate statistics are not necessarily meaningful [3], though we agree that it might provide a more robust idea of model performance, and hence we will report aggregate results in the next draft.
>
> We thank the reviewer for additional feedback to improve the quality of our writing.
>
> [1] Johannes Ballé, David Minnen, Saurabh Singh, Sung Jin Hwang, and Nick Johnston. Variational image compression with a scale hyperprior. ICLR 2018.
> [2] Lucas Theis et al.  Lossy Image Compression with Compressive Autoencoders. ICLR 2017
> [3]  Johannes Ball´e, Valero Laparra, and Eero P Simoncelli. End-to-end optimized image compression. ICLR 2016.

---

> > ### Comment · AnonReviewer3 · 2019-11-15
> > **Response to rebuttal**
> >
> > Thank you for the rebuttal.
> >
> > We agree that even though the competing method [1] uses quantization, it does not seem to suffer from it. You then argue that the other reason for removing quantization is that it circumvents the restriction for uniform distributions of the posterior, and that therefore your method could potentially surpass the results of [1]. If this is the motivation for your work, then you should demonstrate that by using flexible posteriors in your method, you actually improve upon the results of [1] or come close to it. This is currently not demonstrated in the paper, and I therefore retain my score.

---

### Official Review · AnonReviewer1 · 2019-10-24
**Official Blind Review #1**

**Rating:** 3

**Review:**

The authors propose a new image compression method that does not require quantizing the encoded bits in an auto-encoding style image compression model. The method builds on a VAE with image x, code z, and posterior q(z | x). Instead of directly encoding z, the proposed method samples z_{c^\star} from posterior and store c^\star as the compressed representation. A decoder then reconstruct an approximation of x from z_{c^\star}. The authors show that this framework is bits-back efficient and draw connections to prior theoretical results. Experiments were conducted on the Kodak dataset based on the model of Balle et al. The proposed method works only slightly worse than Balle et al. at low bit-rate region, but the gap becomes larger in higher bit rate regions.

The method is technically sound and the paper is clearly written. My main concerns fall in practical aspects. In Figure 3, for 3 out of 4 images, the theoretical upper bound of the proposed method still do not outperform Balle et al. This suggests limitations of the proposed method. Discussion on the limitations of the method is limited. The results also beg the question: How much does quantization in existing methods impact performance, and how much will fixing this benefit the overall system. Finally, in my opinions, in some sense the sampling of z_{c^\star} is also a form of quantization. Does drawing different samples from posterior leads to different reconstructed images? If it does, doesn't it also suffer from similar limitations as existing "quantization" methods?

Overall, I think the direction of the proposed method has good potential, but it also leaves important questions unanswered. I think this paper will benefit from additional revisions.

**Experience Assessment:**

I have published one or two papers in this area.

**Review Assessment: Checking Correctness Of Derivations And Theory:**

I assessed the sensibility of the derivations and theory.

**Review Assessment: Checking Correctness Of Experiments:**

I carefully checked the experiments.

**Review Assessment: Thoroughness In Paper Reading:**

I read the paper at least twice and used my best judgement in assessing the paper.

---

> ### Author Response · Authors · 2019-11-12
> **Responses to Reviewer #1**
>
> We thank the reviewer for their comments on our work. We appreciate the thoughtful feedback.
>
> We agree that the discussion on the limitations of the method is perhaps a bit too limited and could be expanded so that the current challenges of applying our method are clearer. Indeed, even the theoretical upper bound falls off compared to Balle's results. Note that we use Gaussian priors on both levels of our VAE, in particular, standard Gaussians on the second stochastic level, that are not adjusted for the dataset. This is in contrast to Balle's approach, where they utilize a very flexible non-parametric prior for each dimension for their second stochastic level, and separately from the model, they also optimize it for the dataset. We conjecture that the reason this does not adversely affect their test performance is due to the large dataset used for training (~ 1 million high-resolution images).
>
> The extent to which quantization (or the various relaxations of quantization for training) is an interesting question. Figure 4 in [1] shows a nice comparison of the actual quantization error versus the continuous estimate using uniform dither, as well as the approximation quality of the differential entropy used in training with the actual compression rate. It shows that both relaxations are very accurate, the main limitation of their method is that they are constrained to VAEs where the posteriors are always shifted uniform distributions, whereas our method allows the use of arbitrary posteriors.
>
> We do not necessarily agree that the importance sampling algorithm is a form of quantization. Quantization is (usually) a rounding operation, used to assign non-zero mass to the symbols we wish to compress. The importance sampling procedure, on the other hand, serving as part of the relative entropy coding scheme, is lossless.
>
> Drawing different samples from the posterior distribution will naturally cause some variation in the output of the (deterministic) decoder, but VAEs have been demonstrated to be robust to noise on their stochastic level [2], and thus if the difference is not beyond floating-point precision, it certainly is beyond human perceptibility.
>
>
> [1]  Johannes Ball´e, Valero Laparra, and Eero P Simoncelli. End-to-end optimized image compression.
> ICLR 2016.
>
> [2] Bin Dai, Yu Wang, John Aston, Gang Hua, & David Wipf. Connections with robust PCA and the role of emergent sparsity in variational autoencoder models. JMLR 2018

---

### Official Review · AnonReviewer4 · 2019-11-01
**Official Blind Review #4**

**Rating:** 6

**Review:**

This paper studied the image compression problem. Specially, the authors proposed to use neural networks to act as encoder / decoder. The training consists of minimizing the distortion of reconstruction and the difference between approximated posterior and true distribution. The output of encoder is sampled by the proposed relative entropy coding, which extended a previous method by introducing adaptive grouping for acceleration.

In summary, this paper gets rid of commonly used quantization techniques in image compression by using an approximate importance sampler which produces the encoding of images in a non-deterministic manner. With the construction of parameterized encoder / decoder, end-to-end training is conducted by popular gradient descent.

Here is a question:
In experiments, the authors mentioned that the architecture is borrowed from another work. My question is how neural network architecture affects the performance? In other words, how to ensure that the performance is not obtained from the power of the backbone but from the proposed method itself. Are there any possible experiments which can be conducted to show the effectiveness by using different architectures?


**Experience Assessment:**

I do not know much about this area.

**Review Assessment: Checking Correctness Of Derivations And Theory:**

N/A

**Review Assessment: Checking Correctness Of Experiments:**

I assessed the sensibility of the experiments.

**Review Assessment: Thoroughness In Paper Reading:**

I read the paper at least twice and used my best judgement in assessing the paper.

---

> ### Author Response · Authors · 2019-11-12
> **Responses to Reviewer #4**
>
> Thank you for your feedback on our paper.
>
> Our contribution is the coding scheme, and how it can be used in conjunction with generative models. Thus, our main focus was not to find a good VAE architecture, hence we adopted the architecture of [1].
>
> We understand that the choice of architecture in our setting is crucial to good performance, and we chose it precisely to be able to compare the performance of the appropriate trade-offs we made compared to the trade-offs of [1]. Concretely, note that we have usual Gaussian latent space on both levels of our network, which we use for relative entropy coding, whereas [1] have uniform posteriors on both levels, a uniform-Gaussian convolution as the first-level prior and a non-parametric prior on the second,
> that is optimized for the data separately.
>
> Perhaps the most obvious comparison that was left out would be with [2]. The issue is with comparability, namely that [2] trained on a custom dataset and their code was not available so that it could be retrained on the same dataset we trained our model on.
>
>
> [1] Johannes Ball´e et al. Variational image compression with a scale hyperprior. ICLR 2018.
>
> [2] Lucas Theis et al.  Lossy Image Compression with Compressive Autoencoders. ICLR 2017

---

### Official Review · AnonReviewer5 · 2019-11-03
**Official Blind Review #5**

**Rating:** 1

**Review:**

The paper proposes a method for lossy image compression. Based on the encoder-decoder framework, it replaces the discrete codes by continuous ones, so that the learning can be performed in an end-to-end way.

Overall, I think the current version should not be accepted.

The detailed comments are as follows.

1. The novelty is not clear. Using continuous latent space has been analyzed for years. So does the negative beta-ELBO . Section 4 starts with importance sampling, then some implementation compromises, which makes the efficacy of resultant method unclear.

2. You mention that REC achieves bits-back efficiency according to (Hinton 1993). However, how is c* selected in their paper? Now that you use importance sampling, does this still hold?

3. The experiments are very insufficient. It only compares with one method, which is not enough. From the main content, the proposed method is improved upon (Havasi 2018). But it is not compared.  Let alone the other methods mentioned in the related works. PNG and JPEG should also be compared.

4. For the reconstruction, it is better to measure the quality quantitively by PSNR, and mark the Bits/Pixel.

5. Ablation study is also needed. PLN without your contributed part should be evaluated alone. So far, I cannot tell how your method works and which part works.

6. There are many broken sentences and typos.

7. Some statements such as the application parts should be properly cited.

Last, the paper may be entitled as 'image compression without quantization' instead.

**Experience Assessment:**

I do not know much about this area.

**Review Assessment: Checking Correctness Of Derivations And Theory:**

I did not assess the derivations or theory.

**Review Assessment: Checking Correctness Of Experiments:**

I carefully checked the experiments.

**Review Assessment: Thoroughness In Paper Reading:**

I read the paper at least twice and used my best judgement in assessing the paper.

---

> ### Author Response · Authors · 2019-11-12
> **Responses to Reviewer #5**
>
> We thank the reviewer for providing feedback on our paper, we address each point below:
>
> 1. The reviewer is right, continuous latent spaces and using the beta-ELBO as training objective have been studied extensively. The novelty of our work is rather in the compression part, where we show how continuous latent distributions could be used for lossless compression of latent variables, as part of a lossy compression pipeline.
>
> Our work is in contrast with all previous neural compression methods, which all used probability masses and entropy coding.
>
> Second, we also adapt the REC algorithm developed by [1] for BNN compression, to compress the latents of generative models. These adaptations are necessary since i) the structure BNN weights' posterior distribution will differ from the structure of a datapoint's posterior in a generative model, and ii) [1] make use of successive retraining during the compression process, which is far too expensive for a compression codec.
>
> 2. In the work of [2] only the possibility of such a compression method is demonstrated. In our work, we provide a realization of this using our importance sampler.
>
> 3. We agree that more experiments could be performed, e.g. with [3], and across multiple media, e.g. audio or video compression as well. The principal reason for the lack of more experiments was the issue of comparability: there does not seem to be a clear consensus for what training set to use, hence for best comparison we sought to report methods where we could train on the same data as we used for our model and found that of the relevant sources only [4] had their code publicly available.
>
> As mentioned earlier [1] have developed their method for BNN compression, but it does not extend to the lossy data compression setting.
>
> Comparison to PNG is not quite relevant in our setting, as it is a lossless image compression algorithm and we work in the lossy setting.
>
> We compare the performance of our method with JPEG in Figures 3, 7, 8 and 9 on various images from the Kodak dataset.
>
> 4. We show PSNR comparisons in the above-mentioned figures, as well as MS-SSIM,  plotted against the compression rate measured in bits per pixel.
>
> 5. Our architecture is the one used in [4], our contribution is the compression of the latent variables.
>
> 6-7. We would appreciate it if you were able to provide concrete examples where changes are needed.
>
>
> [1] Marton Havasi, Robert Peharz, and Jos´e Miguel Hern´andez-Lobato. Minimal random code learning:
> Getting bits back from compressed model parameters. NIPS workshop on Compact Deep Neural
> Networks with industrial application, 2018.
>
> [2] Geoffrey Hinton and Drew Van Camp. Keeping neural networks simple by minimizing the description
> length of the weights. In Proc. of the 6th Ann. ACM Conf. on Computational Learning
> Theory. Citeseer, 1993.
>
> [3] Lucas Theis et al.  Lossy Image Compression with Compressive Autoencoders. ICLR 2017
>
> [4] Johannes Ball´e, David Minnen, Saurabh Singh, Sung Jin Hwang, and Nick Johnston. Variational
> image compression with a scale hyperprior. In International Conference on Learning Representations,
> 2018.

---

### Decision · Program_Chairs · 2019-12-19

**Decision:**

Reject

**Comment:**

The paper proposes a method for lossy image compression. Based on the encoder-decoder framework, it replaces the discrete codes by continuous ones, so that the learning can be performed in an end-to-end way. The idea is interesting, but the motivation is based on a quantization "problem" that the authors show no evidence the competing method is actually suffering from. It is thus unclear how much does quantization in existing methods impact performance, and how much will fixing this benefit the overall system. Also, the authors may add some discussions on whether the proposed sampling of z_{c^\star} is indeed also a form of quantization.

Experimental results are not convincing. The proposed method is only compared with one method. While it works only slightly worse at low bit-rate region, the gap becomes larger in higher bit rate regions. Another major concern is that the encoding time is significantly longer. Ablation study is also needed. Finally, the writing can be improved.